# Sasso Pisano Geothermal Field Environment Harbours Diverse *Ktedonobacteria* Representatives and Illustrates Habitat-Specific Adaptations

**DOI:** 10.3390/microorganisms9071402

**Published:** 2021-06-29

**Authors:** Sania Arif, Corinna Willenberg, Annika Dreyer, Heiko Nacke, Michael Hoppert

**Affiliations:** 1Department of General Microbiology, Institute of Microbiology and Genetics, University of Göttingen, Grisebachstraße 8, 37077 Göttingen, Germany; corinna.willenberg@stud.uni-goettingen.de (C.W.); annika.dreyer@stud.uni-goettingen.de (A.D.); mhopper@gwdg.de (M.H.); 2Department of Genomic and Applied Microbiology, Institute of Microbiology and Genetics, University of Göttingen, Grisebachstraße 8, 37077 Göttingen, Germany; hnacke@gwdg.de

**Keywords:** Sasso Pisano, hot vents, fumarole, 16S rRNA gene, 18S rRNA gene, microbial diversity

## Abstract

The hydrothermal steam environment of Sasso Pisano (Italy) was selected to investigate the associated microbial community and its metabolic potential. In this context, 16S and 18S rRNA gene partial sequences of thermophilic prokaryotes and eukaryotes inhabiting hot springs and fumaroles as well as mesophilic microbes colonising soil and water were analysed by high-throughput amplicon sequencing. The eukaryotic and prokaryotic communities from hot environments clearly differ from reference microbial communities of colder soil sites, though *Ktedonobacteria* showed high abundances in various hot spring samples and a few soil samples. This indicates that the hydrothermal steam environments of Sasso Pisano represent not only a vast reservoir of thermophilic but also mesophilic members of this *Chloroflexi* class. Metabolic functional profiling revealed that the hot spring microbiome exhibits a higher capability to utilise methane and aromatic compounds and is more diverse in its sulphur and nitrogen metabolism than the mesophilic soil microbial consortium. In addition, heavy metal resistance-conferring genes were significantly more abundant in the hot spring microbiome. The eukaryotic diversity at a fumarole indicated high abundances of primary producers (unicellular red algae: *Cyanidiales*), consumers (*Arthropoda: Collembola* sp.), and endoparasite *Apicomplexa* (*Gregarina* sp.), which helps to hypothesise a simplified food web at this hot and extremely nutrient-deprived acidic environment.

## 1. Introduction

The superheated steams and fumaroles near Sasso Pisano village (Castelnuovo Val di Cecina, Italy) are the central part of the Larderello geothermal field, located in the inner Northern Apennines (Southern Tuscany). The field exhibits extremely high subsurface temperatures (300–350 °C) at a pressure range of 4–7 MPa [1,2,3]. A vapour-dominated reservoir within a metamorphic basement at a depth of 3000 m is connected to a productive 1000 m deep carbonate-evaporite reservoir, which is almost sealed by overlying units. Many artificial steam wells were established for geothermal energy generation. The fumarolic fields of Sasso Pisano between Lagoni del Sasso and Monterotondo Marittimo represent a unique landscape of natural steam vents [3]. The overpressurised hot steam and springs release and/or redeposit CH_4_, CO_2_, sulphur compounds (H_2_S, SO_4_^2−^), boric acid (H_3_BO_3_), ammonia (NH_4_^+^), and heavy metals at the surface [4,5], as water vapours flow (50 t/h) along fault zones/fractures through the sedimentary layers [3,6]. The condensation of sulphur dioxide-rich steam at the surface around fumaroles contributes to an acidic environment (pH 0.5–5) [7], which also affects the surrounding area [8,9].

The Sasso Pisano geothermal field comprises very extreme environmental constraints including heated surface soil (50 to 100 °C), extremely low pH values, low content of bioavailable nitrogen, carbon and phosphorus compounds, and high content of heavy metals as well as trace elements, especially around fumaroles [5,9,10]. These stringent conditions would limit the growth and productivity of microbes and select only a few of the most adaptive extremophiles to survive in the extreme geothermal fields. The bacterial communities of this extreme environment are barely explored, though the extremophilic microorganisms are of great interest due to their potential biotechnological importance [11]. To learn more about the bacterial and eukaryotic communities in the presence of hot steams, environmental DNA from the geothermal fields in Sasso Pisano was isolated for 16S and 18S rRNA gene sequencing and subsequent diversity analysis. The study of the Sasso Pisano fumaroles’ rock and soil samples revealed high predominance of extremophilic novel uncultured *Ktedonobacteria* and eukaryotic *Cyanidiales*. Furthermore, the adaptive strategies of *Ktedonobacteria* and *Cyanidiales* in relation to environmental constraints were inspected. We also analysed if the diversity of *Ktedonobacteria* at Sasso Pisano is similar to that determined in other environments.

## 2. Materials and Methods

### 2.1. Sample Collection

The samples were taken at Sasso Pisano (Pisa, Tuscany Region, Italy) in April 2019 (43°09.632′ North, 10°51.538′ East) and stored at −20° until further use. For prokaryotic community analysis, the sampling sites at the hot springs and nearby soil were sampled (see sample designations in Figure 1). The designation of the samples was SP (Sasso Pisano). The hot springs varied greatly in intensity and discharged whitish to blackish tarnish, accompanied by steam bubbling. Sample SP1 was taken from a greenish/brownish biofilm growing at the top of the hot spring A vent, constantly exposed to steam. Sample SP2 was taken from a yellowish/beige, possibly elemental sulphur deposit formed by vapours at the hot spring A. SP11 (greyish muddy watery discharge) and SP12 (yellowish greyish sediments) were directly sampled from the hot spring B vent. The hot spring C was sampled at different distances from the openings SP24, SP26 and SP7. The hot spring D was sampled at the vent (SP310) and when the discharge was meeting hot spring C sediments and organic material (plant leaves: SP38) (Figure 1D). A description of the features and details with respect to all samples are provided in Appendix A. Soil samples (Sp513–515, Sp3, Sp517, Sp719, Sp819–821, Sp122) with various textures were frequently collected around the different hot springs. To study a thermophilic eukaryotic consortium, growing under extremely acidic (pH 2) and hot conditions (44–55 °C), a greenish biofilm (SP5) was collected from the vertical walls of a small gorge filled with hot steam seeping out of the ground (Figure 1H). For comparison among the extreme sites, a water stream with a neutral pH and a lower temperature (15 °C) near the fumarole crossing the site was sampled and labelled as SP4 (control).

### 2.2. Environmental DNA Extraction, Amplicon Generation and Sequencing

The Power soil DNA isolation kit (Qiagen, Hilden, Germany) was used to extract the whole genomic DNA of the stored samples. The samples were subjected to three extractions (triplicates) each, according to the manufacturer’s protocol. The extracted gDNA was then concentrated with the Eppendorf Concentrator Plus (Eppendorf, Hamburg, Germany) at 45 or 60 °C for 35 min to obtain the optimal gDNA concentration. Qualitative and quantitative determination of DNA was performed by gel electrophoresis and a Nanodrop instrument (NanoDrop 2000 c Spectrophotometer, ThermoFisher Scientific, Waltham, MA, USA). After quality assessment, partial 16S and 18S rRNA genes were amplified via polymerase chain reaction (PCR). For eukaryotes, the 18S amplicon PCR forward primer = 5′-TCG TCG GCA GCG TCA GAT GTG TAT AAG AGA CAG CCA GCA SCY GCG GTA ATT CC-3′ and the reverse primer = 5′-GTC TCG TGG GCT CGG AGA TGT GTA TAA GAG ACA GAC TTT CGT TCT TGA TYR A-3′ were used. For bacteria, the 16S amplicon forward primer 5′-TCG TCG GCA GCG TCA GAT GTG TAT AAG AGA CAG CCT ACG GGN GGC WGC AG-3′ and the reverse primer = 5′-GTC TCG TGG GCT CGG AGA TGT GTA TAA GAG ACA GGA CTA CHV GGG TAT CTA ATCC-3′ were applied [12,13]. These primers were additionally linked to the overhang adapter sequences to make the amplicon suitable for Illumina MiSeq sequencing. Both partial 16S and 18S rRNA genes were amplified via a modified PCR Master mixture: Double-distilled nuclease-free water (32 μL) was mixed with 10 μL Phusion GC Buffer-5×, 10 μM forward and reverse primers (1.0 μL each), 5% DMSO (2.5 μL), 50 mM MgCl_2_ (0.15 μL), 10 mM dNTPs (1.0 μL), 25 ng template DNA (2.0 μL) and 0.5 μL of 2 U/μL Phusion HF DNA polymerase (ThermoFisher Scientific) to obtain a final volume of 50 μL reaction mixture. A thermal cycler (Biometra, Göttingen, Germany) was used to anneal the 18S and 16S rRNA gene-specific primers with the extracted gDNA at 56 and 60 °C, respectively. PCR was performed as described [13,14]. The amplification products were qualitatively and quantitatively controlled on a 0.8% agarose gel with 1× TAE buffer (ThermoFisher Scientific) and a NanoDrop spectrophotometer (ThermoFisher Scientific). PCR products were further processed by using the GeneRead Size Selection Kit (Qiagen) to wash off primers and PCR reagents from the resulting amplicons. Indexing of OCR products was executed with the Nextera XT DNA library prep kit (Illumina, San Diego, CA, USA); subsequently, paired-end sequencing at the Göttingen Genomics Laboratory using the 2 × 300 bp Paired-End mode with an Illumina MiSeq sequencer (Illumina) was performed. The obtained Fastq sequences are accessible at the NCBI database under the Sasso Pisano microbiome project number PRJNA725822.

### 2.3. Processing of 16S rRNA Gene-Based Amplicon Sequencing Data

The MetaAmp automated pipeline, an online resource for metagenomic analysis, was utilised to analyse 16S rRNA gene-based amplicon sequencing data (http://ebg.ucalgary.ca/metaamp/, accessed on 18 November 2020) [15]. Initially, the USEARCH software merged the uncompressed and demultiplexed sequence data and removed low-quality reads [16]. The subsequent step was the removal of non-matching and misaligned read pairs of small lengths (less than 350 bp). Moreover, the Mothur software package was utilised for the identification and trimming of forward and reverse primers [17]. Any reads that lacked forward and reverse primers or containing primer mismatches were discarded. In the next step, the reads were dereplicated using UPARSE software version 7.1 [18]. This software also discarded chimaeras and singletons. The resulting, high-quality reads were assembled into taxonomic units (OTUs) based on the 0.97 OTU clustering threshold (97% identity). The Mothur software package assigned taxonomic status to the OTUs with reference to the SILVA training dataset (http://www.mothur.org/wiki/Taxonomy_outline, accessed on 18 November 2020). Lastly, the aforementioned software generated relative abundance data, rarefaction curves, as well as alpha and beta diversity indexes. The samples were rarefied into subsamples. The lack of similarity within different samples was computed through the Bray–Curtis index and principal coordinate analysis (PCoA) was performed. The Permutational Multivariate Analysis of Variance (PERMANOVA) test allowed for hypothesis testing.

### 2.4. Analysis of 18S rRNA Gene-Based Amplicon Sequencing Data

With respect to analysis of 18S rRNA gene-based data, paired-end reads were merged using PEAR (Paired-End reAd mergeR) [19] and resulting sequences of small lengths (less than 400 bp) were removed. Furthermore, the BBMap tool was used to remove all sequences that did not match the primer sequences at the 5′ and 3′ ends [20]. Subsequently, chimeric sequences were removed with the VSEARCH tool [21]. Data cleaned in this way were converted from the FASTQ file format to the FASTA format and mapped with QIIME (Quantitative Insights into Microbial Ecology) against the reference genes from the SILVA 132 18S database [22] with a 97% match [23].

### 2.5. Transmission Electron Microscopy (TEM)

For TEM sample preparation, the samples SP4 and SP5 were placed in a fixing solution (0.2%, *v*/*v*, glutardialdehyde, 0.3%, *w*/*v*, formaldehyde). Subsequently, they were washed with a 50 mM K_2_HPO_4_ buffer solution and parts of the biofilm were embedded in agar (Agar-Agar, Kobe I, Carl Roth, Karlsruhe, Germany). Sliced small agar blocks of samples were dehydrated with ethanol in increasing concentration, followed by infiltration with synthetic resin (LR White Resin, London Resin Company, Berkshire, UK), and then, embedded in gelatine capsules, which were polymerised at 55 °C for 24 h.

To obtain ultra-thin sections, the polymerised samples were first pre-milled with a sample trimming device (TM 60, Reichert-Jung, Wetzlar, Germany), and sliced with the ultramicrotome (Ultracut E, Reichert-Jung) using a diamond knife (DuPont, Wilmington, DE, USA). The quality of the ultrathin sections was assessed based on their interference pattern and “fished” out of the water using coated grids (Plano, Wetzlar, Germany) as specimen support. For staining of ultrathin sections, uranyl acetate and replacement stain (UAR-EMS Uranyl Acetate Replacement Stain, Electron Microscopy Sciences, Hatfield, UK) were applied at different dilutions (undiluted, 1:3, 1:5, *v*/*v*). The grid with the section side facing downwards was placed on a drop of 20 μL of the respective staining agent and was incubated for ten minutes. The ultrathin sections were examined by transmission electron microscopy (JEM-1011 Electron Microscopes, JEOL, Akishima, Tokyo, Japan) at a voltage of 80 kV and electron micrographs were recorded.

### 2.6. Prokaryotic Community Functional Profiling

The MetaAmp-generated taxonomic profile data were utilised to generate a holistic overview of functional profiles of various samples. In this context, the Piphilin online server (http://secondgenome.com/Piphillin, accessed on 5 October 2020) [24] was used. Piphilin allows for the direct search of representative OTU sequences against a database composed of 16S rRNA gene sequences using USEARCH version 8.0.1623 [16]. In the next step, gene copy numbers of every inferred genome are summed to generate KO abundances (the KEGG reference database was used) [25]. The online available server MicrobiomeAnalyst [26] was utilised for the completion of statistical and meta-analysis of gene abundance data. The R package DESeq2 was used to compute a differential abundance analysis [27]. To summarise the findings, important features of hot sampling sites were considered in regard to soil samples that exhibited over a 1-fold difference.

### 2.7. Phylogenetic Analysis

The phylogenetic analysis was focused on 16S rRNA gene sequences, including those of different *Ktedonobacteria* type strains. MUSCLE, implemented in MEGA-X software version 7.0, was utilised to align these sequences and also *Ktedonobacteria* class-related OTU sequences [28]. Lastly, as part of the phylogenetic analyses, evolutionary distances were assessed. The Kimura 2-parameter model was used for this. The construction of phylogenetic trees was performed utilising the maximum likelihood method [29]. Bootstrap values were calculated based on 1000 replications.

## 3. Results

### 3.1. Prokaryotic Community Composition

A total of 788,808 high-quality 16S rRNA gene sequences were available for analysis of prokaryotic communities. Per sample, the sequence amount ranged from 14,219 to 92,423 and a total of 1797 OTUs were identified (Appendix A). The sampling sites at the hydrothermal field were divided into two groups: hot springs and soils. The relative abundances at hot springs varied greatly per sample, emphasising each hot spring microbiome is unique (Figure 2). For instance, the prokaryotic profile of the samples SP24 and SP26, emerging from the same blackish muddy hot spring C, was dominated mainly by *Aquificae* (*p* = 0.001083), and hot spring B SP11 and SP12 samples were dominated by *Deinococcus*/*Thermus* (*p* = 1.8017 × 10^−15^). *Chloroflexi* were abundant in the SP1 biofilm taken from hot spring A expelling clear water steam fumes, and samples SP11 and SP12 from slightly muddy water hot spring B. The phylum-level profile also changed drastically when the hot spring streams (SP24, SP26 and SP310) were coming into contact with soil (SP27) or other organic matter (leaves, SP38) downstream, suggesting the integration of a mesophilic consortium (including *Firmicutes* or *Proteobacteria*). The overall composition of the hot spring samples comprised *Chloroflexi* (21%), *Proteobacteria* (19%), *Firmicutes* (17%), *Cyanobacteria* (8%), *Aquificae* (6%), *Acidobacteria* (5%), *Deinococcus Thermus* (4%), *Euryarchaeota* (3%), *Thermotogae* (6%) and *Actinobacteria* (2%). In contrast, in all soil samples, the most abundant taxa were *Cyanobacteria* (27%), *Proteobacteria* (19%), *Acidobacteria* (11%), *Actinobacteria* (10%), *Chloroflexi* (8%), *Euryarchaeota* (7%), *Planctomycetes* (6%) and *Thaumarchaeota* (4%) and the composition showed no drastic variations among most of the samples. An increased abundance of *Chloroflexi* was detected in soil samples that were collected nearby fumaroles or hot springs.

At the class level, the differences between the hot spring samples were more prominent. For instance, the SP1 biofilm from hot spring A showed a high abundance of the branched hyphae and spore forming *Ktedonobacteria* (67%), while the other hot spring sample SP12 was enriched with filamentous thermophilic *Anaerolineae* (36%). *Deinococci* (27%), *Aquificae* (54–50%), *Gammaproteobacteria* (21–32%)*, Acidobacteria* (16%), and *Oxyphotobacteria* (54%) were the most abundant taxa in samples SP11, SP24, SP26, SP27 and SP310, respectively. *Bacilli* (*Firmicutes*) inhabited the surface of yellowish sulphur deposits (44%, SP2) and a downstream site (65%, SP38) where the whitish discharge mixed with leaves and blackish muddy discharge from hot spring C. In the group of soil samples, the abundant classes were, in different samples, *Oxyphotobacteria*, *Thermoplasmata*, *Alphaproteobacteria* and *Gammaproteobacteria*, *Acidobacteria* and *Actinobacteria*. The depicted pie charts show taxa in the hot spring and soil group in the order of ascending relative abundance (Figure 3). Overall, the hot spring group showed a higher relative abundance of thermophilic taxa.

### 3.2. Diversity of Ktedonobacteria

The 30 most abundant OTU sequences potentially affiliated with *Ktedonobacteria* were aligned with 16S rRNA gene sequences of isolated *Ktedonobacteria* strains and metagenomic sequences resulting from NCBI blast. In the phylogenetic tree, the OTUs are sandwiched between mesophilic (*Dictyobacter*) and thermophilic (*Thermogemmatispora*) groups of the *Ktedonobacteria*, indicating the presence of a large pool of uncultured mesophilic to thermophilic *Ktedonobacteria* strains. Most strains could be classified as members of *Ktedonobacteraceae*; however, OTUs related to JG30-KF-AS9 and B10-SB3A seem to be outliers and are more closely related to thermophilic *Thermogemmatisporaceae* (Figure 4). The detected OTUs indicate that the natural hydrothermal field of Sasso Pisano offers a rich reservoir of novel uncultured *Ktedonobacteria*, which should be targeted with respect to cultivation, genome sequencing, and further exploration. The highest abundance of *Ktedonobacteraceae* was detected in the SP1 biofilm, which indicates a natural enrichment of these bacteria at a hot spring vent (Appendix A). More diverse genus members belonging to *Ktedonobacteria* were identified in another hot spring sample (SP310; whitish discharge).

### 3.3. Alpha and Beta Diversity

For alpha and beta diversity analysis, sequence datasets were rarefied to the lowest detected read size per sample. The Shannon index indicated that some of the hot spring samples comprise a lower diversity than soil group samples (Figure 5). The soil group potentially exhibited a higher alpha diversity in some cases, as the environmental conditions are comparably less harsh and facilitate a broad range of microorganisms to grow. In contrast, due to the acidic pH, higher temperature, and constant washout at the hot spring spots, only acidic thermophilic microbes are supposed to grow. However, the *t*-test result does not show a significant difference between the two groups, indicating that the soil microbial communities may provide an active microbial influx to the hot springs and vice versa (Figure 5A).

To visualize and explore the complex metagenomic data, the multidimensional scaling method—Principal Coordinates Analysis (PCoA)—was applied to observe the similarities at the OTU level. The PCoA of Sasso Pisano samples showed that the microbial communities of hot springs and soil samples were clustering separately, with a considerable variance among the two groups (Figure 5B). According to the Permutational Multivariate Analysis of Variance (PERMANOVA), the beta diversity among both groups also suggested the inhabitant microbial communities are distinct, indicating the stringent effect of hot spring constraints in selection and enrichment of a unique microbial consortium as compared to the soil microflora. The tree diagram, calculated based on a Bray–Curtis index, also indicated that both groups are clearly distinct (Figure 5C).

### 3.4. Functional Profile

The abundance of functional genes summarised by KO identifiers (also called K numbers) was estimated from the OTU data. Based upon this estimation, the predicted function of genes was used to evaluate a metabolic profile of the microbial communities in different sites. The differential abundance analysis of KO numbers indicated that the hot spring microbiota was metabolically rich in terms of heavy metal resistance, methane, sulphur and nitrogen metabolism as well as aromatic compounds’ degradation pathways as compared to the soil microbial consortium. Numerous transporters involved in the extrusion of Mn, Zn, Cu, W, Co, Ni and Mo were present in the hot spring microbial consortium (Table 1). The functional profile also showed that the different KO numbers associated with methane metabolism are differentially present in hot spring and soil samples (Figure 6). This indicates that the hot spring microbiome utilises different enzymes as compared to the soil microbiome to metabolise environmental methane. The soil microbiome shows preferences for aerobic methane metabolism such as methane oxidation to formaldehyde and formate dehydrogenation to CO_2_, while the hot spring microbiome seems to favour anaerobic methane metabolism through reverse-methanogenesis by means of methyl-coenzyme M reductase (Mcr-AOM) [30,31] (Appendix A). Moreover, with respect to carbon fixation, mainly formaldehyde seems to be fixed via the ribulose-monophosphate pathway (RuMP) [32]. More KO numbers in the hot spring microbial communities were also observed to be differentially abundant, involved in sulphur and nitrogen metabolism (data not shown). In the aromatic compound degradation pathways, genes and enzymes involved in the aminobenzoate, benzoate, chlorocyclohexane, chlorobenzene, chloroalkane, chloroalkene, dioxin, fluorobenzoate, nitrotoluene, naphthalene, styrene, xylene and toluene degradation were also differentially abundant in the hot spring samples (Appendix A).

### 3.5. Eukaryotic Diversity at a Fumarole

According to rarefaction curves, 300–450 OTUs were identified in the neutral pH and mesophilic stream conditions; the extreme fumarole site comprised less than 50 OTUs (Appendix A). A total of 57 OTUs affiliated with known taxonomic groups were identified in stream sample (SP4) data, while the fumarole rock (SP5) data comprised a total of 21 groups, which shows a considerably higher alpha diversity in the water stream biome than in the fumarole biome. It can be concluded that the extreme conditions of the fumarole limit microbial alpha diversity, while microbial communities could easily populate the neutral pH water stream.

Organisms known to typically colonise freshwater habitats, including amoebae (11.2%) which can be divided into the groups *Dactylopodida* (1.1%), *Filamoeba* (7%) and *Ischnamoeba* sp. (2.3%), were detected. The *Incerta Sedis* represents a kind of placeholder for unspecified species in the genome database. The paraphyletic group of green algae is represented by *Chlorophyta* (0.75%) and *Charophyta* (4%). Among the multicellular organisms, *Chaetonotida* (10%), belonging to the phylum *Gastrotricha* (“hairybacks”), was detected. Furthermore, *Adineta vaga*, belonging to the rotifers, was found with 2.1% and the lower fungi were represented with a total of 5.7% relative abundance. The group of SAR (*Stramenopiles*, *Alveolata*, *Rhizaria*) showed 19.7% and LEMD267 is listed in SILVA as a nonspecific taxonomic group that refers to lobose amoebae (22.5%). However, this cannot be further determined with the available data. Excluding the unspecified eukaryotes (19%), 62.02% of the identified species OTUs (*Amoeba* 22.5%, *Archaeplastida* 0.82%, *Metazoa* 14.3%, *SAR* 19.7%, low mushrooms 4.7%) can be assigned to a biofilm in a freshwater area with a neutral pH and a lower temperature around 15 °C. The other major abundant taxa were *Intramacronucleata* (Ciliates) (16%) and *Chaetonotida* (10%). The OTU assigned to *Rhodophyceae* with 0.07% and *Echinamoeba thermarum* with 4.2% can be potentially explained by the association of the stream channel to the thermo-acidic sampling site, since both OTUs possess a thermophilic character.

The apparently qualitative identical species composition of the sample duplicates (SP4.1 and SP4.2) differs especially in terms of the fraction of nematode- and mite-related OTUs (Figure 7). The unspecified *Cyanidiaceae* and the species *Galdieria sulphuraria* belonging to the family *Cyanidiales* of phototrophic eukaryotic algae represent the most abundant taxa of the fumarole biofilm (65.3%). Unspecified *Collembola* (springtails) and *Folsomia candida* contributed to *Collembola* sp. (16.8%). *Collembola* sp. (*Arthropoda*: *Hexapoda*) represent more highly developed organisms, as they are among the typical bottom dwellers and have been described for other extreme sites, in particularly hot and particularly cold areas [33]. *Gregarina caledia* (*Apicomplexa*) contributed 14% of the biofilm consortium (Figure 7).

### 3.6. Structural and Morphological Description of a Eukaryotic Biofilm

The electron micrographs of a fumarole biofilm (SP5) preparation allowed inspection of individual cells, which are separated from each other by a cell wall of approx. 0.2 μm thickness. Within these cells, organelles, chloroplasts, in particular, are visible (Figure 8). Inside chloroplasts, membrane stacks of the thylakoids are prominent features. Densely packed cells (Figure 8A) are separated from each other by an extracellular matrix formed between the organisms. Remineralisation processes have caused dark precipitates to settle as small particles in the extracellular matrix. Larger particles are separated from the cells by the matrix (Figure 8A). The marked particles imitate the shape of the matrix.

Based on the number of spore cells within the mother cell, potentially either the genus *Galdieria* or *Cyanidium* could be identified. More than four spores in the cell represent a distinct structural feature of *Galdieria* (Figure 8B), while a sporangium with tetraspores could be classified as *Cyanidium* (Figure 8C). Based on the species description in Ciniglia et al. (2004) [34], the organism (Figure 8D,E) can be morphologically determined even more precisely as *Galdieria sulphuraria* under the electron microscope. The cell has a thick cell wall and contains at least seven visible chloroplast shapes in the cytosol, surrounding a central vacuole, visible as a membrane-bounded empty lumen. Ultrathin sections also reveal the coexistence of the bacterial colony outside the extracellular matrix of the *Cyanidiaceae* biofilm (Figure 8F).

## 4. Discussion

*Ktedonobacteria*, a deeply branched bacterial class comprising of mesophilic and thermophilic representatives, is characterised by its ubiquitous prevalence in terrestrial environments, complex life cycle, and in some cases, large genome size [35]. The class is divided into two orders, *Ktedonobacterales* and *Thermogemmatisporales*. The latter order includes *Thermosporotrichaceae* as well as *Thermogemmatisporaceae* strains, whereas *Ktedonobacterales* comprise *Dictyobacteraceae, Ktedonobacteraceae* and *Ktedonosporobacteraceae* strains. Within these two orders, 20 proposed mesophilic and thermophilic species have been assigned to the genera *Dictyobacter*, *Tengunoibacter*, *Ktedonobacter*, *Ktedonosporobacter*, *Thermosporothrix* and *Thermogemmatispora* [36,37,38,39]. *Ktedonobacteria* isolates and related environmental DNA are derived from non-extreme sources [38,39] and extreme environments, such as an acid vapour-formed spring [40], naturally occurring CO_2_ gas vents [41], a lava cave in a volcanic trench [42], volcanic fumaroles [43], steaming geothermal soil [44], and a mineral precipitating cave environment [45]. This suggests that *Ktedonobacteria* members appear to prevail in oligotrophic and extreme environments, implying these strains may have evolved versatile metabolic pathways to cope with extreme conditions. For instance, the potential to oxidise carbon monoxide (CO) has been reported for *Ktedonobacteria* members [46].

In this study, we gained insights into the abundance and diversity of *Ktedonobacteria* in the geologically diverse and environmentally extreme hydrothermal field environments of Sasso Pisano. Microbial diversity varied greatly with the type of sampling site, which included hot springs, a fumarole, and nearby soil as well as water samples. A high abundance and diversity of *Ktedonobacteria* at the hot springs and in some soil samples indicated that the Sasso Pisano hydrothermal region offers a natural reservoir of *Ktedonobacteria* members, which became naturally enriched under stringent environmental constraints including heated surface soil, extremely low pH values, low nutritional content of nitrogen, carbon and phosphorus and high content of heavy metals as well as trace elements, especially around fumaroles [5,9,10]. The functional profile indicated that the hot spring microbiome includes a significantly higher abundance of organisms harbouring genes involved in methane, sulphur, nitrogen and aromatic compounds’ metabolism, since the acidic hot springs and fumaroles emit a mixture of various gases such as ammonia, methane, carbon dioxide, hydrogen, hydrogen sulphide, hydrocarbons and aromatic compounds [2,5,47]. Moreover, transporters conferring resistance against toxic transition metals were also detected in the hot spring microbiome, which implies that *Ktedonobacteria* members inhabiting this site could have evolved high metabolic plasticity to cope with acidification, nutrient depletion, and heavy metal resistance. In our previous study of biofilm samples from Marsberg copper mine, Germany, the capability to degrade toxic aromatic compounds as well as resistance against transition metals was detected in a metagenome-assembled genome affiliated with *Ktedonobacteria* [48]. The bacterial biofilm on steam vents from Sapichu volcanoes was also mainly colonised by related similar taxa of *Chloroflexi* (*Ktedonobacteria*), *Acidobacteria,* and *Cyanobacteria*. The metabolic potential analysis predicted similar cellular metabolic pathways related to methanogenesis, sulphur respiration, nitrogen fixation, and heavy metal transport except for photosynthesis by *Cyanobacteria* [43]. The observed higher abundance of *Cyanobacteria* in low-temperature soil samples (but not in hot springs samples) was consistent with colonisation patterns of *Cyanobacteria* in lower temperature vents, as reported by Wall et al. [49], while the higher temperature vents were abundant in *Cloroflexi* [49,50].

Two important metabolic pathways in oligotrophic geothermal sites are attributed to trace gas (CO and H_2_) utilisation by the type I carbon monoxide dehydrogenase (*cox* genes) and the [NiFe]-hydrogenase (*hyp* genes) as an energy source for cell growth and persistence under nutrient-limiting conditions [51]. Mounting genomic evidence suggests that three phyla, *Chloroflexi*, *Actinobacteria* and *Acidobacteria*, use CO and H_2_ as substrates [52]. A variety of thermophilic bacteria belonging to the classes *Actinobacteria*, *Deinococci, Ktedonobacteria, Thermomicrobia* and *Clostridia* contain *cox* operons and may be capable of aerobic CO oxidation. Hot springs from different regions host different *cox* encoding communities [53]. The *Ktedonobacteria* genome also includes the reductive TCA cycle along with multiple copies of *cox* operons, conferring CO oxidation potential to this taxa, allowing it to predominate the microbial community [46] under the influence of CO-rich gas vents, hydrothermal springs, and soil environments [41,54,55]. Other high abundant taxa have evolved diverse strategies, for instance, most *Anaerolineaceae* species metabolise various organic carbon sources under anaerobic conditions through fermentative metabolism [56]. Representatives of the thermophilic *Aquificae* grow in hot springs via oxidation of dissolved ferrous iron or iron-containing minerals and conduct nitrogen fixation even at 70 °C [57,58]. *Deinococci* are resistant to several stresses due to their highly efficient DNA damage repair ability and detoxification of several toxic compounds through hydrolytic activity [59,60]. Some *Oxyphotobacteria* (photosynthetic *Cyanobacteria*) are metabolically diverse primary producers and pioneer the colonisation where light is available including ecosystems with low light, low levels of O_2_ and/or sulfidic conditions, because of their ability to perform, besides oxygenic photosynthesis, anoxygenic photosynthesis [61,62].

The eukaryotic alpha diversity at the fumarole was significantly lower than in the river water sample, whereas the alpha diversity of prokaryotes did not vary significantly in the hot springs and soil samples. This indicates that thermophilic eukaryotic organisms are less diverse than thermophilic prokaryotes. *Cyanidiales* dominated the biofilm with an OTU of 65.3%, as they represent the only phototrophic organisms growing under these conditions and have been extensively studied since the 1980s [63]. The *Cyanidiales* form a monophyletic group within the red algae, which inhabits the acidic hot springs at different sites. *Cyanidiales* were identified under the electron microscope based on the chloroplasts and reproduction pattern, as the genera *Cyanidium* and *Galdieria*. *Cyanidium* sp. propagate with tetraspores, whereas *Galdieria* is larger and forms more autospores and vacuoles than *Cyanidium* [64]. Yoon et al. (2006) described the structural composition of endolithic and interlithic populations of *Cyanidiales* at Pisciarelli in the Phlegraean fields near Naples, in Sasso Pisano (distinct from the site sampled in this study) and Monte Rotondo [65]. Overall, *Galdieria* dominated the fumaroles of Sasso Pisano, which was expected due to the mixotrophy, the resulting adaptation to fluctuating environmental factors, and availability of carbon to be exploited. *Cyanidium caldarium* was also identified.

Current descriptions of the organisms around geothermally active areas around the world usually only illuminate the photoautotrophic community of the order *Cyanidiales* without considering the heterotrophic eukaryotes. The investigation of the entire diversity is interesting as the extreme conditions can serve as models for living conditions in the early evolutionary stages of soil ecosystems and can provide further information on the development of biocoenoses in the history of the earth. On the other hand, they represent a reduced biocoenosis that can serve as a model for more complex ecosystems. *Collembola* sp. (*Hexapoda*) represent the largest and most complex species at this site. *Collembola* are among the typical bottom dwellers and have been described for other extreme sites, in particularly hot and cold areas [34]. They are known as consumers of bacteria, algae, and fungi [66,67,68], actively feeding and thriving on these biomass sources. *Gregarina* sp. are known to endoparasitise *Hexapoda* among invertebrates in numerous colonies [69]. Based on the detected taxa, a simple food web could be assumed in which *Cyanidium* and *Galdieria*, belonging to unicellular red algae *Cyanidiales*, represent the primary producers and build biomass which may be consumed by *Collembola* sp. as consumers, and *Gregarina* sp. endoparasitise *Collembola* sp. Symbiotic bacteria may facilitate the biofilm growth of *Cyanidiales. Collembola* sp. could alter the microbial communities, either directly (through selectively feeding) or indirectly (through the dissemination of microbial propagules). Ultimately, a more detailed description of the species for this site would be required to confirm the described food web as a next step. This can then serve as a model to describe the influences and interactions of the organisms involved on each other and relate these to more complex ecosystems.

## 5. Conclusions

The composition and morphology of microbiota around hot fluid springs, steam vent and soil samples from the geothermally active hydrothermal fields around the Sasso Pisano site (Pisa, Tuscany region) are presented. The prokaryotic community analysis indicated that *Ktedonobacteria* at the Sasso Pisano site ranged from thermophilic to mesophilic, and phylogenetically as well as metabolically diverse representatives were detected. Metabolic functional profiling of the thermophilic prokaryotic microbiota predicted a higher capability to utilise carbon (methane and aromatic compounds), sulphur and nitrogen compounds, and heavy metal resistance-conferring genes were also significantly more abundant in the hot spring microbiome as compared to the mesophilic soil microbial consortium. The diversity of eukaryotic microorganisms within biofilms at extreme conditions, with very high temperature (55 °C) and a very low pH value of 2, was evaluated. The detection of red algae *Cyanidiales*, *Arthropoda* (*Collembola* sp.) and *Apicomplexa* (*Gregarina* sp.) proposed a simplified food web at the thermophilic extremely nutrient-deprived acidic environment of a fumarole.

## Figures and Tables

**Figure 1 microorganisms-09-01402-f001:**
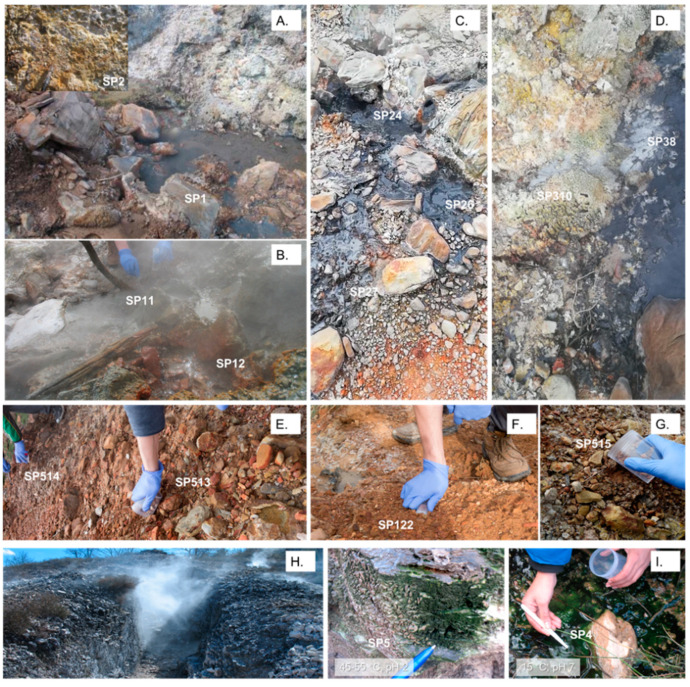
Sampling sites for 16S rRNA and 18S rRNA gene-based analysis. (**A**–**D**) are hot springs sites, while (**E**–**G**) are representatives of the soil sampling sites. Each sample is labelled “SP” (Sasso Pisano), followed by numbers. The picture inserted in (**A**) represents the yellowish-orange SP2 sampling site, formed by the surface deposition of the sulphur vapours probably from a solfatara-type vent. Hot spring sites (**A**,**D**) were either of clear or whitish liquid discharge. Hot springs (**B**,**C**) were splashing muddy greyish/blackish water. The fumarole rocky wall was continuously wrapped in hot steam (**H**). The greenish biofilm forms a tight layer on the crumbly rock of the fumarole (SP5) and was compared with a cold and pH-neutral water stream (SP4) (**I**).

**Figure 2 microorganisms-09-01402-f002:**
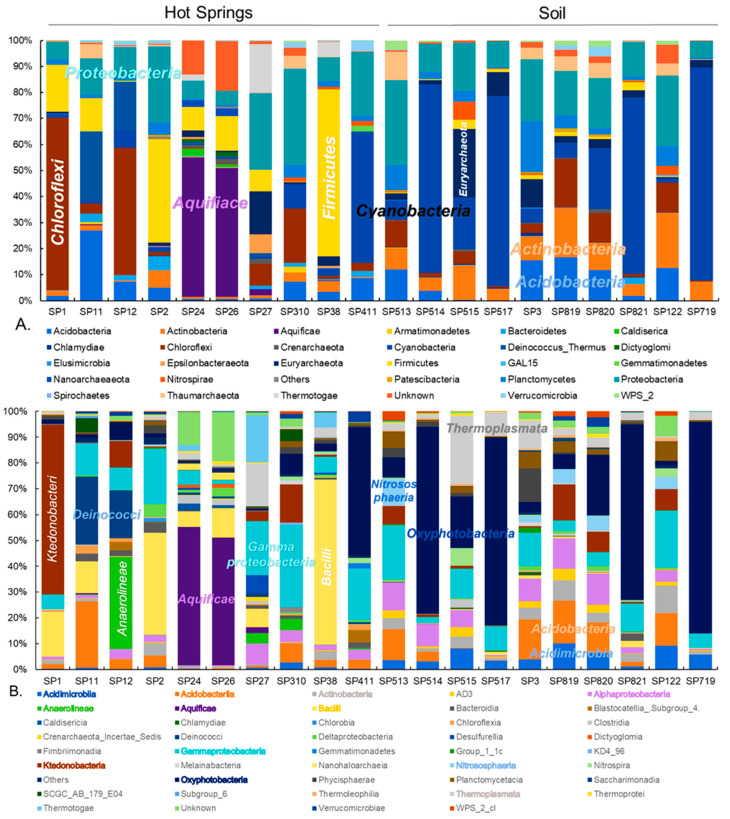
Relative abundance of prokaryotic taxa in Sasso Pisano hot spring and soil samples. Prokaryotic community composition at the phylum level (**A**) and class level (**B**) is shown for individual samples. Abundant taxa are highlighted within stacked bars or within taxa legends.

**Figure 3 microorganisms-09-01402-f003:**
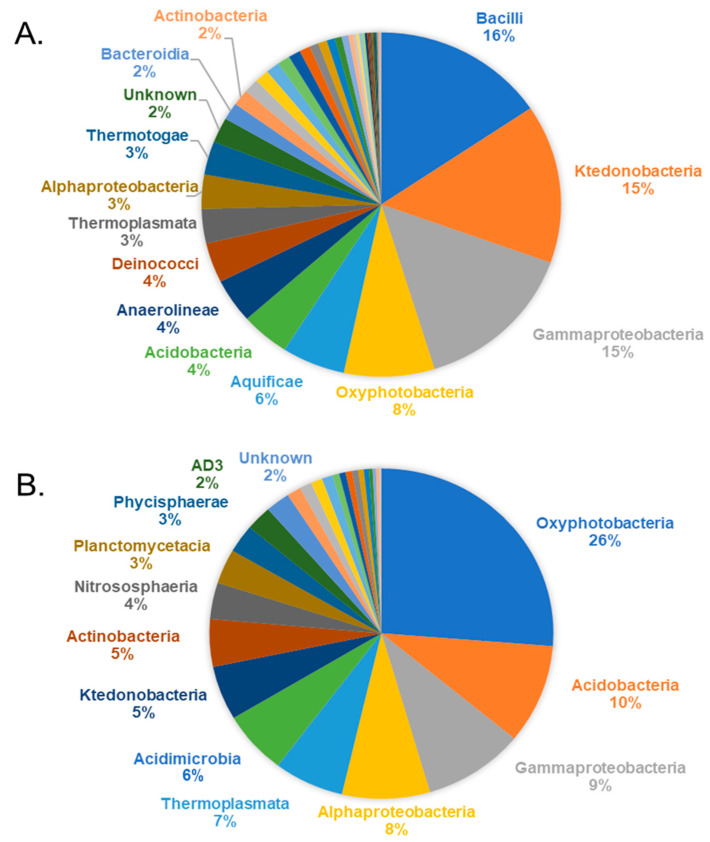
Overall bacterial community composition of the hot spring and soil microbiome. The actual abundances at the class level are plotted for the hot spring (**A**) and soil group (**B**). Taxa showing less than 2% relative abundance are not displayed.

**Figure 4 microorganisms-09-01402-f004:**
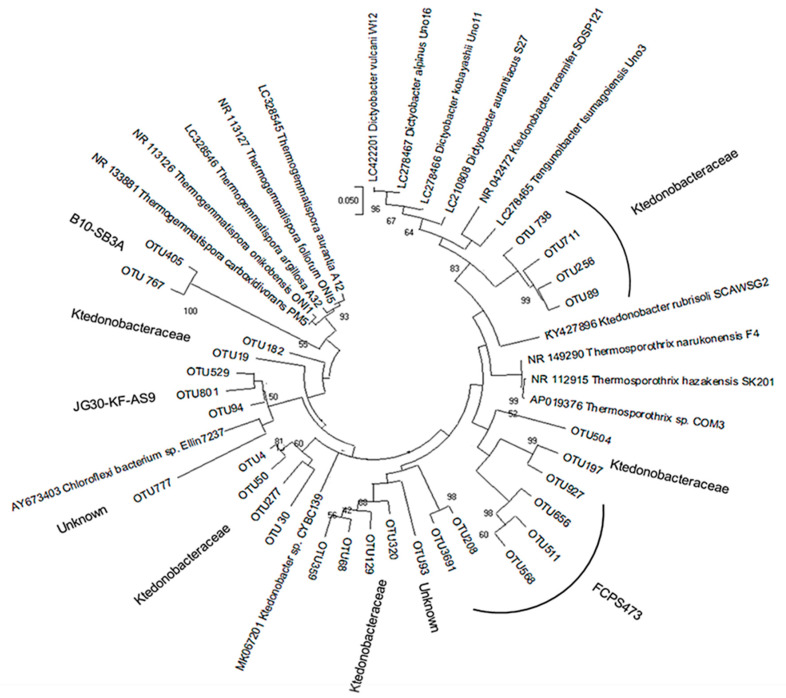
Maximum likelihood phylogenetic tree including *Ktedonobacteria* OTUs. The phylogenetic tree was constructed by aligning OTU sequences potentially affiliated with *Ktedonobacteria* with 16S rRNA gene sequences of known *Ktedonobacteria* species.

**Figure 5 microorganisms-09-01402-f005:**
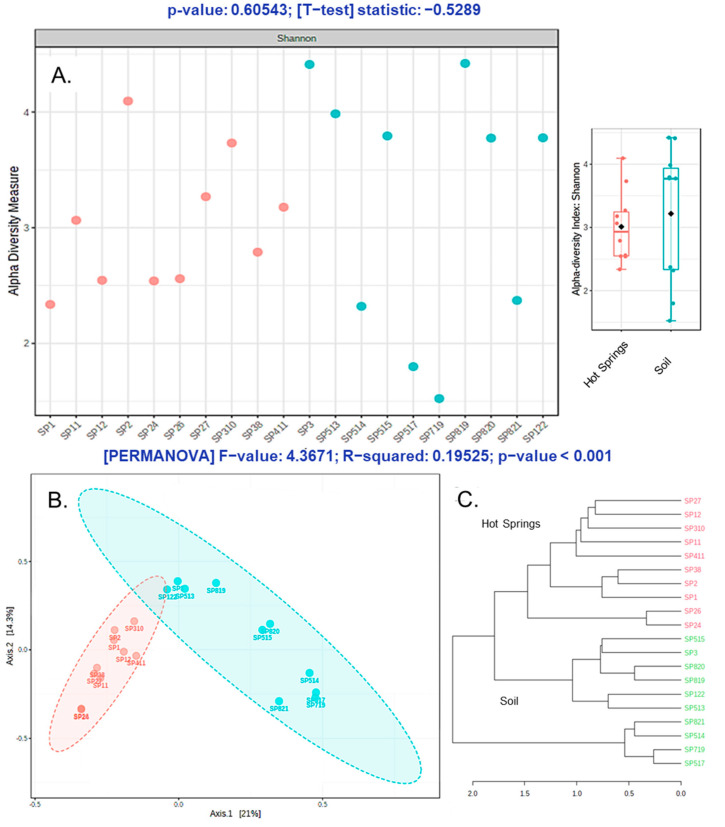
Comparative alpha and beta diversity analysis of the hot springs versus soil samples. The Shannon index (**A**) showed that both hot springs and soil samples were rich in terms of microbial alpha diversity, though the particular microbial composition for both habitats was distinct, based on the PCoA (**B**) and Bray-Curtis tree (**C**) analysis.

**Figure 6 microorganisms-09-01402-f006:**
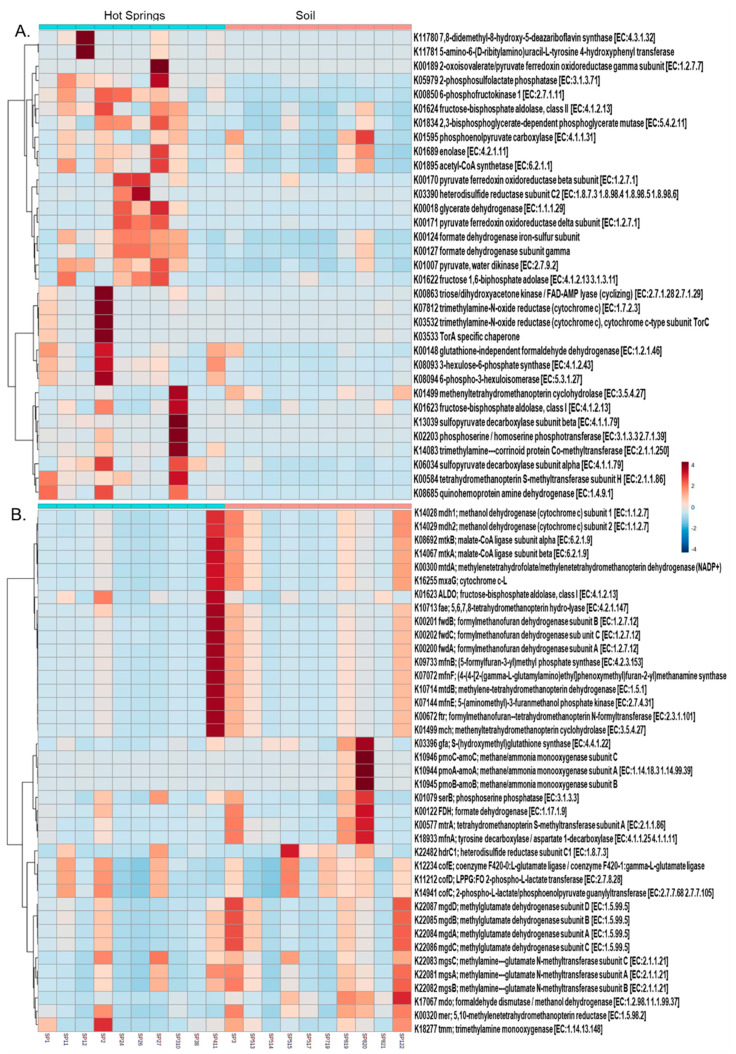
Differentially abundant genes involved in methane metabolism. The hot spring microbiome (**A**) was found to be abundant in the genes involved in the anaerobic methane degradation as compared to the aerobic methane degradation in the soil microbiome (**B**).

**Figure 7 microorganisms-09-01402-f007:**
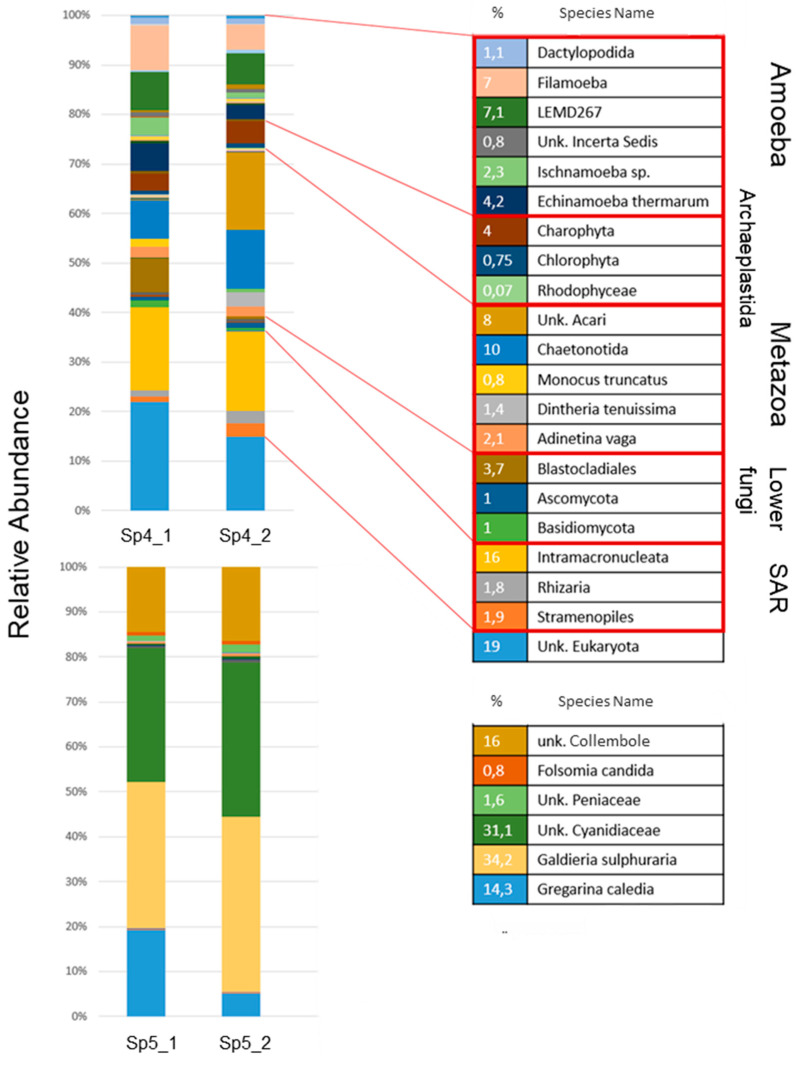
Eukaryotic community composition of a neutral pH (water stream biofilm SP4) and extreme (fumarole biofilm SP5) biome. Taxonomic groups are listed which show at least 0.80% of the total OTU abundance (except for the *Rhodophyceae* (0.07%) from the stream water sample to show the contrast to the fumarole and *Chlorophyta* to distinguish them from *Charophyta*). In the stream water sample, 21 species groups are listed which represent 98.02% abundance of the total OTUs, and six groups are shown in the fumarole samples, which represent 97.90% of the total OTUs (Unk, unknown).

**Figure 8 microorganisms-09-01402-f008:**
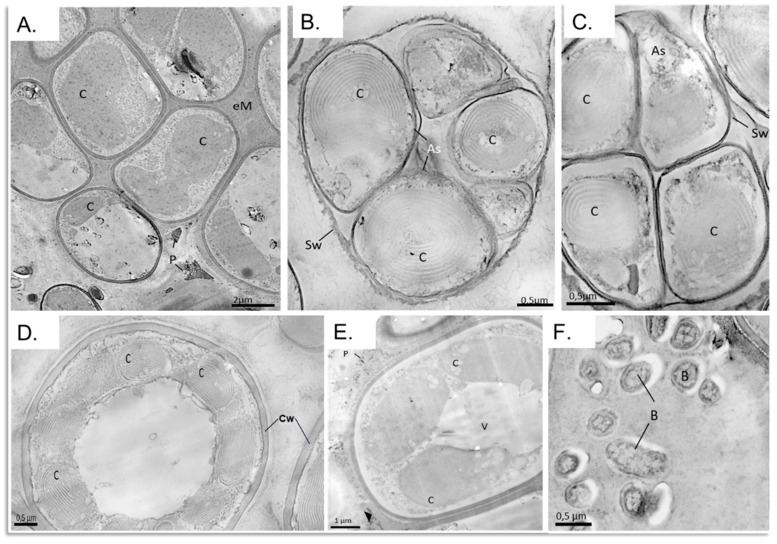
Electron microscopic analysis of a section through a sample taken from the fumarole biofilm SP5. An overview of a *Cyanidiales* cell aggregate (**A**). Sporangia (mother cells; **B**,**C**) containing a varying number of autospores. In three of the five autospores (**B**), the membrane stacks of the thylakoids in the chloroplasts are clearly visible. Within the mother cell (**C**), four autospores (tetraspores) already are about the same size and the sporangium wall is disintegrating. The cells in (**D**,**E**) carry multiple chloroplasts around a central vacuole. Both cells could be assigned to the genus *Galdieria*. Prokaryote morphotypes of about 0.5 μm in size attached to the *Cyanidiales* aggregate (**F**). V, vacuole; eM, extracellular matrix; Cw, cell wall; Sw, sporangium wall; B, bacteria; As, autospores; P, precipitates.

**Table 1 microorganisms-09-01402-t001:** Differentially abundant transporters required for heavy metal resistance in hot springs microbiome.

Transport System	
**Manganese**	**Zinc**
K19973 mntA; manganese ATP-binding protein K19975 mntC; manganese substrate-binding protein K19976 mntB; manganese permease protein K11601 mntC; manganese substrate-binding protein K11603 mntA; manganese ATP-binding protein K11602 mntB; manganese permease protein	K11707 troA; manganese/zinc/iron substrate-binding protein K11708 troC; manganese/zinc/iron permease protein K11709 troD; manganese/zinc/iron permease protein K11710 troB; manganese/zinc/iron ATP- binding protein
**Iron**	**Copper**
K11604 sitA; manganese/iron substrate-binding protein K11605 sitC; manganese/iron permease protein K11606 sitD; manganese/iron permease protein K11607 sitB; manganese/iron ATP-binding protein K02010 afuC; iron(III) ATP-binding protein K02011 afuB; iron(III) permease protein K02012 afuA; iron(III) substrate-binding protein	K19340 nosF; Cu-processing system ATP-binding protein K19341 nosY; Cu-processing system permease protein
	**Tungstate**
	K05772 tupA; tungstate substrate-binding protein K05773 tupB; tungstate permease protein
**Molybdate**	**Nickel**
K02017 modC; molybdate ATP-binding protein K02018 modB; molybdate permease protein K02020 modA; molybdate substrate-binding protein	K15584 nikA; nickel substrate-binding protein K15585 nikB; nickel permease protein K15586 nikC; nickel permease protein K15587 nikD; nickel ATP-binding protein
**Sulphate/Thiosulphate**	**Cobalt**
K02045 cysA; sulfate/thiosulfate ATP-binding protein K02046 cysU; sulfate/thiosulfate permease protein K02047 cysW; sulfate/thiosulfate permease protein K02048 cysP; sulfate/thiosulfate substrate-binding protein	K02006 cbiO; cobalt/nickel ATP-binding protein K02007 cbiM; cobalt/nickel permease protein K02008 cbiQ; cobalt/nickel permease protein K02009 cbiN; cobalt/nickel transport protein

## Data Availability

The amplicon sequencing data presented here is accessible at the NCBI database under the Sasso Pisano microbiome project number PRJNA725822.

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
