# Peer review of "Sasso Pisano Geothermal Field Environment Harbours Diverse Ktedonobacteria Representatives and Illustrates Habitat-Specific Adaptations"

_microorganisms, 2021, doi:10.3390/microorganisms9071402_

Round 1
Reviewer 1 Report
The quality of the work is significantly improved, there are some stylistic imperfections in the text (such as bold in the discussion part), but all in all it is a good job
Author Response
The authors would like to thank you the reviewer as it helped to significantly improve the manuscript and authors also proofread the manuscript again to make necessary corrections.
Reviewer 2 Report
The authors have addressed my comments to the former submission. The resubmitted manuscript has been improved although still contains some typos, errors, and unclear sentences, including:
Abstract (L. 11): Please correct "Pisano".
l. 443-444: The sentence "Since Cyanobacteria were abundant in our lower temperature soil samples, not in hot springs samples." is not clear. Please correct.
l. 501, 502, 503, 521, 522: “sp.” should not be in italics.
l. 456-457: “which gives this taxon the potential of CO oxidation through to predominate the microbial community” is unclear.
Author Response
The authors acknowledge and appreciate the constructive suggestions of the reviewer and made the changes as mentioned.
Abstract (L. 11): Please correct "Pisano".
>corrected
l. 443-444: The sentence "Since Cyanobacteria were abundant in our lower temperature soil samples, not in hot springs samples." is not clear. Please correct.
>corrected in line 451-455
l. 501, 502, 503, 521, 522: “sp.” should not be in italics.
> sp. style changed to not italics
l. 456-457: “which gives this taxon the potential of CO oxidation through to predominate the microbial community” is unclear.
> corrected in line 466-467
This manuscript is a resubmission of an earlier submission. The following is a list of the peer review reports and author responses from that submission.
Round 1
Reviewer 1 Report
This research is devoted to the abundance of prokaryotes and eukaryotes in hydrothermal environments of Sasso Pisano, Italy, as well as their metabolic potential and resistance to stress factors. The functional profile of the community is rather clear, as well as the microbial diversity. However, the roles of particular microbial taxa identified by the authors in the communities and these environments, especially Ktedonobacteria, were not clearly defined in the text of the manuscript. The metabolic roles of the predominant members of the studied communities should be discussed.
- 86: Please add (I.) to the description of Figure 1. One subfigure is not designated.
The manuscript contains some errors. For example,
- 41: “thereby reaching…” is unclear in this sentence. Please correct.
- 395: Please correct “size”.
- 396: Why is the designation of “cell wall” “Zw”?
Reviewer 2 Report
In this work, the authors describe the microbial communities that live in the environments of Sasso Pisano, Italy.
I suggest several improvements:
The abstract requires general improvements, with particular attention in the final part and I suggest to use a different word instead of a reservoir.
Also, in the introduction, the word reservoir is used too often.
Table S1 is missing
I suggest summarizing the characteristics of the sampling sites (also adding the temperature and pH) in a table to be included in the main text and not as supplementary material.
I suggest reducing the details on the commands used “In the Processing of 16S rRNA gene-based amplicon sequencing data.”
Supplementary figures are missing at all.
The name of the identified species must be corrected in italics
I suggest to remove the name of the species from the graph and leave them in the legend of figure 1. The names on the columns are not read clearly.
Figure 6: I think this figure needs to be re-designed, it's not clear at all
Figure 7 is not legible on the right side, I suggest increasing its quality